# Traditional Fermented Foods from Ecuador: A Review with a Focus on Microbial Diversity

**DOI:** 10.3390/foods11131854

**Published:** 2022-06-23

**Authors:** Luis Santiago Guerra, Juan Manuel Cevallos-Cevallos, Stefan Weckx, Jenny Ruales

**Affiliations:** 1Department of Food Science and Biotechnology, Escuela Politécnica Nacional, P.O. Box 17-01-2759, Quito 170517, Ecuador; luis.guerra@epn.edu.ec; 2Centro de Investigaciones Biotecnologicas del Ecuador (CIBE), Campus Gustavo Galindo, Escuela Superior Politécnica del Litoral (ESPOL), Km 30.5 Vía Perimetral, P.O. Box 09-01-5863, Guayaquil 090112, Ecuador; jmceva@espol.edu.ec; 3Research Group of Industrial Microbiology and Food Biotechnology (IMDO), Faculty of Sciences and Bioengineering Sciences, Vrije Universiteit Brussel (VUB), Pleinlaan 2, B-1050 Brussels, Belgium; stefan.weckx@vub.be

**Keywords:** traditional foods, microbial fermentation, cacao, coffee, chicha, champús, starter culture

## Abstract

The development of early civilizations was greatly associated with populations’ ability to exploit natural resources. The development of methods for food preservation was one of the pillars for the economy of early societies. In Ecuador, food fermentation significantly contributed to social advances and fermented foods were considered exclusive to the elite or for religious ceremonies. With the advancement of the scientific research on bioprocesses, together with the implementation of novel sequencing tools for the accurate identification of microorganisms, potential health benefits and the formation of flavor and aroma compounds in fermented foods are progressively being described. This review focuses on describing traditional fermented foods from Ecuador, including cacao and coffee as well as less popular fermented foods. It is important to provide new knowledge associated with nutritional and health benefits of the traditional fermented foods.

## 1. Introduction

Food fermentation is being used in different parts of the world as a method for food preservation and to improve the sensory attributes of food. Early civilizations used fermentation as a method for food preservation of vegetables, fruits, and diverse types of meat. Food fermentation has prompted the development of new products, which in many cases contributed to the advancement of the social, political, economic, and cultural aspects of society [1]. The food fermentation process consists of a series of biochemical reactions that—when occurring correctly—can improve the food digestibility and bioavailability of nutrients [2], enhance the organoleptic characteristics [3] and increase the quality of foods [4,5]. In addition, fermented foods can improve gastrointestinal health [6,7], lower blood cholesterol and pressure levels, and have shown anti-carcinogenic and anti-inflammatory activity [8,9,10]. Furthermore, some fermented-food microorganisms are known to also promote gastrointestinal health and inhibit intestinal pathogens [11].

Food fermentation is a complex biochemical process, in which microorganisms play an important role for the development of sensorial attributes [12]. In general, microorganisms break down different substrates present in the food matrix into important compounds that add value to food products using a wide range of different enzymes [13]. Indeed, during the fermentation process, carbohydrates present in the raw food matrices are usually converted into ethanol and carbon dioxide by yeasts as well as some heterofermentative lactic acid bacteria, whereas acetic acid bacteria convert ethanol mainly into organic acids, such as acetic acid. All these chemical conversions usually cause a decrease in the pH of the matrix, which benefits food preservation [14].

There is a growing demand for healthy food products, including those resulting from fermentation processes [15]. In addition, the number of scientific, peer-reviewed papers about fermented foods has increased in 5.99% since 2014 (PubMed, 2021). Additionally, the interest in research about the benefits, production, and prospects of fermented foods is increasing over the years. Regarding market share, in 2018, the annual sales of yogurt was USD 110 million in the United Stated, representing a compound annual growth rate (CARG) of 6.3 % during 2022–2027 [16]. Similarly, the value of the global kombucha market was approximately USD 1.5 billion in 2018 [17] and the annual sales of the global vinegar market reached USD 2.27 billion in 2021 [18]. Similarly, the global kefir market size was 1.23 billion in 2019 and could increase to 1.84 billion by 2027 [19]. Therefore, the fermented foods market has been proposed as a tool to target sustainable development by the year 2030 and to fight hunger and malnutrition. To achieve this, legislation and research-based policies have been suggested as a mechanism to support the implementation of microbial biotechnology in industrial set-ups [20]. 

## 2. Fermented Foods from Ecuador

The spontaneous fermentation of food and beverages has traditionally been applied by many cultures in South America. In Ecuador, fermentation processes have influenced the local gastronomy with a significant number of fermented foods consumed for generations, including cacao, coffee, and fermented beverages, such as chicha, chaguarmishqui and champús. This review focuses on the microbiological composition of the most important Ecuadorian fermented foods.

### 2.1. Cacao

Fermented cacao beans are the main raw material used to produce chocolate. The high worldwide demand for chocolate contributed to the fact that 1.8 million hectares were dedicated to cacao cultivation in Latin America in 2019 [21], with Ecuador being the top fine-flavor cacao-producing country worldwide. Only in the first five months of 2021, Ecuadorian exports of fermented cacao reached USD 266.4 million. Recent genomic research and archeological evidence showed that the domestication of the *Theobroma cacao* plant potentially occurred in the humid Amazon forest [22].

Chocolate production starts with the cacao fermentation process. Cacao fermentation is mostly carried out by different types of microbial communities, including yeast, lactic acid bacteria (LAB), and acetic acid bacteria (AAB) in cacao beans [23]. Overall, yeast usually act at the beginning of the fermentation process to degrade carbohydrates, depolymerize pectin, and produce ethanol [24]. At these early stages, pH and temperature changes occur [25], so ethanol production is usually accompanied by an increase in temperature from 35 to 40 °C within 48 h from the beginning of the fermentation [26]. *Saccharomyces, Hanseniaspora*, and *Pichia* are usually the most abundant yeasts found at the beginning of the fermentation process, although they are found at a lower abundances at later stages of the cacao fermentation process [27,28,29,30].

During the initial phases of cacao fermentation, colonization by Enterobacteriaceae, such as *Tatumella* spp., allows the assimilation of citric acid and production of gluconic acid [31]. Then, LAB perform heterolactic or homolactic fermentation processes to produce lactic acid, acetic acid, ethanol, and mannitol [32,33]. The most common genera of LAB found during the cacao fermentation process are *Lactiplantibacillus*, *Limosilactobacillus*, *Bacillus*, *Fructobacillus*, *Enterococcus*, *Leuconostoc*, *Streptococcus*, and *Weissella* [27]. LAB are Gram-positive and anaerobic bacteria that play an important role in carbohydrate and ethanol degradation. Additionally, LABs are responsible for the production of certain secondary metabolites, such as diacetyl and acetoin. These compounds possess a butter-like and butter scotch flavor, and are widely used as safe flavoring agents and potentially antifungal compounds [34,35]. Additionally, phenolic acid and flavonoids could be converted into other aromatic precursors with synergistic and/or additive effects that might contribute to the overall inhibition of molds [36]. 

In the final stage of the fermentation process, AAB oxidize ethanol to acetic acid and eventually overoxidize the acetic acid into carbon dioxide. Although, the presence of *Acetobacter* spp. in the first stages of fermentation it is not very high, the process allows them to develop and then become very abundant later on [37]. AAB are Gram-negative and aerobic bacteria that can also promote the taste, texture, and smell of fermented products, such as cacao [38,39].

The quality of the fermented cacao beans depends on the cacao variety and various post-harvest processes. Additionally, cacao fermentation may yield batches with different quality as the microbial diversity of cacao can be influenced by the climatic region and farm practices [25,26]. Some studies have estimated various diversity indexes, such as Ace, Chao, Simpson, and Shannon, during cacao fermentation [40].

The diversity indexes have contributed to the analysis of fermented foods and have allowed an understanding of microbial ecosystems in the foods. Table 1 shows a variation in the species richness from different research on cacao and coffee bean fermentation [40]. Cacao and coffee are the most important fermented foods in Ecuador. For both products, microbial fermentation contributes to the hydrolyzation and removal of the pulp. The draining of the pulp facilitates the subsequent drying process. In addition, fermentation in both cacao and coffee contribute to the formation of precursors of aroma compounds. 

The values as Shannon’s index can be affected by initial microbiota, chemical composition of the beans, temperature, and interactions among microorganisms [41]. Shannon values for spontaneous fermentation between 2 and 3 are considered normal.

### 2.2. Coffee

The demand for coffee is also increasing worldwide. In 2018, global purchases of imported coffee increased to USD 32.9 billion, which represents a 9.4% increase when compared to the values of 2016 [43]. Coffee cultivation migrated through various parts of the world before reaching Latin America, where Brazil is currently the most important coffee exporting country. Although the benefits of coffee consumption are still under scrutiny, the energetic and therapeutic effects of coffee have been highlighted [44]. The different sensorial characteristics of coffee are largely dependent on the plant genotype, cultivation region, and the post-harvest processing methods used, including the fermentation process [45].

In general, coffee fermentation starts with the harvest of ripe coffee cherries. Immediately after harvesting, the cherries are de-pulped and fermented in sealed tanks before drying on handmade structures and greenhouses. However, the post-harvest and fermentation processes of coffee are usually specific to each farm, and can be classified as wet, dry, or semi-dry, each one triggering the development of unique characteristics [46]. In the wet method, the cherries are selected and submitted to submerged fermentation for 12 to 36 h to remove the mucilaginous layer. In the dry method, the cherries are dried together with the shell and fermented simultaneously, yielding an early perception of caramel sensation followed by chocolate and fruit-like aromas [47]. In the semi-dry method, the cherries are first de-pulped and then collected to ferment and dry in the open air, typically for 10 to 15 days and the result is usually a specialty coffee with citric and herbaceous flavors, as well as in some cases fruity notes [4]. In all cases, obtaining a high cup quality is influenced by the yeast richness and the abundance of Lactobacillales [42]. The genera *Saccharomyces*, *Pichia*, and *Hanseniaspora* are amongst the predominant yeasts, while *Enterobacter*, *Pantoea*, *Enterobacteriaceae*, and *Rahnella* have been reported as the predominant Gram-negative bacteria. Bacillus has been reported as the predominant Gram-positive bacterial genus [48].

The microbial species richness of fermented coffee from different Ecuadorian farms [49], as well as the impact of the microbial communities and their enzymes in coffee fermentation processes has been assessed [50]. Various microorganisms and a variety of extracellular enzymes contribute to the production of ethanol, acetic acid, and lactic acid. The most important enzymes acting in the coffee fermentation process are pectinase, polygalacturonase, and pectin methyl esterase, all of them using pectin as substrate [50,51]. The microbial diversity found during the coffee fermentation process has been associated with the altitude of the process above sea level (asl). In a study carried out in Brazil, at 800 m asl (meters above sea level), *Gluconobacter* (19.8%), *Novosphingobium* (18.9%), and *Sphingomonas* (12.2%) showed the highest abundances. However, *Weissella* (32.7%), *Sphingomonas* (36.2%), and *Methylobacterium* (39.4%) were the most abundant species at 1000 m asl, 1200 m asl, and 1400 m asl, respectively, with the highest yeast abundance found at 1000 m asl [52,53]. Other environmental and process conditions also influence the microbial community composition and activity [54]. Two strains of *S. cerevisiae*, obtained from different locations, showed clear differences in development and growth [4]. 

Among the different types of coffee, Luwak coffee is listed as the best in the world and is produced by a natural fermentation method in many countries of Southeast Asia, including Philippines, Malaysia, and Indonesia. This type of coffee is fermented in the intestinal tract of wild civet cats where various enzymes and intestinal microorganisms cause changes in the chemical composition of the beans. However, the high price has caused an increased hunting of Luwak animals. For this reason, the use of starter cultures with species of *Gluconobacter*, *Lactobacillus*, *Leuconostoc*, and *Streptococcus* isolated from the intestinal tract of Luwaks, have been proposed [55,56].

### 2.3. Traditional Fermented Beverages 

Fermented beverages, such as chicha and champús, have traditionally been used for religious ceremonies and produced at small scales during specific festivities. During the fermentation process of these beverages, different microbial changes occur according to the geographical region and the methods used at home, village, or community in charge of the preparation. To date, there have been few reports on the microbiota of both drinks, as discussed in the sections below.

#### 2.3.1. Chicha

Chicha is an Andean ancestral beverage that has remained perennial in different culinary rituals for more than 3000 years. Chicha is considered a traditional beverage representing brotherhood and reciprocity. Manufacturing chicha is a domestic and communal activity from Colombia, Ecuador, Peru, Brazil, and Bolivia. In Ecuador, different kinds of chicha can be found, including chicha de jora, chicha de cassava, and chicha de Yamor (also known as seven-grains chicha) [57]. Within ancestral cultures, such as the Incas, chicha consumption represented a high lineage and prestige. Today, chicha is used in ceremonial activities, for instance, to welcome an important person, as an accompanying drink of traditional dishes, and as a refreshing drink during community jobs (also known as mingas) [58,59].

The most popular chicha is chicha de jora, which is made from yellow maize as the main ingredient. The making of chicha de jora starts when maize grains are germinated inside containers with water for 13 days (Figure 1). During this time, enzymes inside the grains break down the starch into simple sugars. Then, grains are separated from the water with a mesh and sun-dried to stop biochemical reactions. The dried grains are ground to obtain a flour that is subsequently mixed with water and transferred into special vessels, known as Pondos, for spontaneous fermentation. Some producers add other ingredients, such as a brown sugar loaf known as panela, herbs and spices [59,60]. Other types of chicha, including “chicha de cassava”, are made in the Amazon region of Ecuador using cassava (*Manihot esculenta*) or chonta (*Bactris gasipaes*). This type of chicha is usually first chewed by the indigenous women and children to mash the cassava and break down the starch by the amylases present in the saliva, while providing saliva microorganisms for fermentation [61]. Similarly, “chicha de Yamor” is prepared with seven different types of maize, and is mostly consumed in the town of Otavalo in Ecuador [59]. Few studies have reported the interactions, relationships, and development of microorganisms during chicha fermentation. Yeasts from the genera *Saccharomyces, Torulaspora, Pichia, Candida*, and others have been reported to consume carbohydrates present and produce ethanol during the chicha fermentation [62]. Among the yeasts, *Saccharomyces cerevisiae* and *Torulaspora delbrueckii* have been the most frequently isolated from chicha samples. Restriction polymorphism mitochondrial DNA (mtDNA) analyses revealed a high diversity of *S. cerevisiae* from chicha, as 68 different mtDNA molecular profiles have been identified among 121 yeast isolates [59]. 

Various LAB, such as *Lactiplantibacillus plantarum* (previously known as *Lactobacillus plantarum)*, *Leuconostoc*, and *Streptococcus*, have contributed to an increased acidification, viscosity, and aroma formation in chicha [11,61,63]. Additionally, species of *Klebsiella*, *Bacillus*, *Staphylococcus*, *Micrococcus*, *Enterobacter*, and *Weissella* were detected in chicha samples from Brazil [64]. However, *Acetobacter* spp. was the only AAB found in chicha de jora from Peru, whereas the genera of potential foodborne pathogens and spoilage microorganisms have been seldomly found in the chicha samples [65]. Currently, using molecular microbiological methods, microorganisms involved in chicha fermentation are investigated to apply in novel fermentation strategies in the food industry [60]. 

#### 2.3.2. Champús

Champús is a traditional beverage from Colombia, Ecuador, and Peru. The production of champús usually starts with grinding different cereals, such as wheat, rye, and maize, to obtain a flour that is then mixed with water. The flour–water mixture is placed in vessels (pondos) for approximately three days to allow microbial fermentation. Additionally, other non-flour ingredients are usually added, such as panela, pineapple, naranjilla (*Solanum quitoense* Lam), chamburo (*Vasconcellea pubescens*), syrup, clove, cinnamon, and orange tree leaves [66]. Finally, a low-alcohol beverage with a sweet-acid taste and a particular aroma is obtained (Figure 2). During the fermentation of the cereals, the presence of yeasts, such as *S. cerevisiae*, *Issatchenkia orientalis*, *Pichia fermentans*, *P. kluyveri* var. *kluyveri*, *Zygosaccharomyces fermentati*, *Torulospora delbrueckii*, *Galactomyces geotrichum*, and *Hanseniaspora* spp., have been reported [67]. Even though it is known that there are different bacterial groups within the fermentation process, there are no reports to date on the identification and isolation of microorganisms from champús.

## 3. Functionality of Microorganisms 

In general, the fermentation of traditional foods is performed by yeast, LAB, AAB, among others (Table 2). Yeasts are one of the main microbial groups responsible for food fermentation, and mainly perform fermentation to obtain energy from carbohydrates, such as maltose, sucrose, glucose, and fructose, to generate ATP [68]. During this biochemical process, the yeasts transform carbohydrates into alcohol. Traditionally, fermenting yeasts can be classified as *Saccharomyces* or non-*Saccharomyces*. *Saccharomyces cerevisiae* has commonly been found in the fermentation processes of various foods, and is the most used in industrial processes because of its fermentative capacity, rapid growth, and easy adaptation [69]. At the same time, the diastatic properties of *S. cerevisiae* have offered insight into mechanisms used for adaptation to fermentation environments and formation of flavor-active esters [70]. Additionally, the interaction between *Saccharomyces* and other yeasts, such as *Starmerella*, *Torulaspora*, *Hanseniaspora*, and *Metschnikowia*, has shown a strong effect in the final carbohydrate and nitrogen contents of the fermenting matrix. This fact could be related to the nature and diversity of secreted proteins during the fermentation process [71] and a faster consumption of glucose, ammonium, and arginine [72].

LAB are Gram-positive cocci or bacillus-shaped, catalase-negative, and aerotolerant microorganisms. LAB can be classified as homofermentative and heterofermentative depending on the metabolic pathways that each species uses for carbohydrate consumption and metabolite production [73,74]. In general, LAB have the ability to metabolize carbohydrates mainly into lactic acid. The most common LAB associated with foods are *Lactococcus*, *Streptococcus*, *Enterococcus*, *Pediococcus*, *Leuconostoc*, *Oenococcus*, *Tetragenococcus*, *Carnobacterium*, *Weissella*, and *Lactobacillus*, although the genus *Lactobacillus* has recently been reclassified into 23 novel genera [74,75]. Some LAB strains, such as *Lactiplantibacillus plantarum* HEAL9, *Lacticaseibacillus rhamnosus 271*, *Weissella confusa* MD1, and *Weissella cibaria* MD2, have shown probiotic properties [76,77]. 

Similarly, AAB are another bacterial group commonly associated with food fermentation. AAB, such as *Acetobacter* and *Gluconobacter* spp., usually carry out acetic acid fermentation by which acetic acid and other volatiles with aroma descriptors of fruits, trees, and chocolate are produced [78].

**Table 2 foods-11-01854-t002:** Microbial species reported in fermented foods.

Community	Species	Fermented Food *	Study Observations	Reference
Yeast	*Candida californica*	C, CJ	The genus Candida is frequently found in spontaneous fermentation processes and has been assessed as a starter culture for alcohol production.	[67]
	*Candida humilis*	CJ, Cf		
	*Candida quercitrusa*	Cf		
	*Candida sake*	CJ, CM		
	*Candida solani*	C, CJ		
	*Candida sorbosivorans*	C		
	*Candida sorboxylosa*	CJ		
	*Candida zeylanoides*	CJ		
	*Candida vinaria*	CJ		
	*Candida tropicalis*	C, CY	Used as a starter culture in sorghum beer and barley malt medium.	[79,80]
	*Dekkera anomala*	CJ	These species were related to the production of unpleasant aromas and were not recommended as starter culture.	[81]
	*Dekkera bruxellensis*	CJ, SC		
	*Hanseniaspora opuntiae*	C, CY	Was used as a possible starter culture in cacao fermentation.	[82]
	*Hanseniaspora uvarum*	Cf	Starter culture for the production of volatile compounds in fermented foods and beverages.	[71,83]
	*Hanseniaspora spp.*	CJ, CH, Cf		
	*Issatchenkia orientalis*	CH	Malic acid reduction and interaction mechanisms with *S. cerevisiae*.	[84]
	*Kazachstania exigua*	CJ	Starter culture for cacao fermentation.	[85]
	*Kodamaea ohmeri*	CY		
	*Kluyveromyces marxianus*	C		
	*Pichia fermentans*	CJ, CM, CH	Starter culture for wine to stabilize color and increase fruit and floral aromas.	[86]
	*Pichia kluyveri*	C, CJ, CH	Starter culture to increases volatile thiols (3-mercaptohexanol and its acetylated derivative 3-mercaptohexyl acetate) with fruity aroma as passion fruit and grapefruit.	[87]
	*Pichia kudriavzevii*	C,	Starter culture for cacao fermentation.	[85]
	*Pichia manshurica*	SC		
	*Rhodotorula minuta*	C		[60]
	*Rhodotorula mucilaginosa*	CJ, Cf		
	*Saccharomyces cerevisiae*	C, Cf, CJ, SC, CY, CH	Starter culture for different types of beer.	[88]
	*Saccharomycodes ludwigii*	CJ, CM		
	*Torulospora delbrueckii*	C, CJ, CM, CY, CH	Alcohol production to improve flavor diversity.	[60,89]
	*Zygoascus hellenicus*	CJ		
	*Zygosaccharomyces fermentati*	CH		
LAB	*Enterococcus casseliflavus*	C	Food bio-preservative.	[90]
	*Enterococcus saccharolyticus*	C		
	*Enterococcus sp.*	C		
	*Fructobacillus durionis*	C	Some strains with probiotic potential.	[49]
	*Fructobacillus ficulneus*	C		
	*Fructobacillus tropaeoli*	C		
	*Lactobacillus acidophilus*	C, CY	Starter cultures for steering food fermentation processes. Some specific strains have probiotic potential.	[91,92].
	*Lactobacillus amylovorus*	C		
	*Levilactobacillus brevis*	C, Cf		
	*Liquorilactobacillus cacaonum*	C		
	*Lacticaseibacillus casei*	C, CJ		
	*Loigolactobacillus coryniformis*	C		
	*Lactobacillus delbrueckii*	C, CY		
	*Lactiplantibacillus fabifermentans*	C		
	*Lentilactobacilluss farraginis*	C		
	*Licmosilactobacillus fermentum*	C, CY		
	*Lactobacillus garvieae*	C		
	*Liquorilactobacillus nagelii*	C		
	*Lactiplantibacillus plantarum*	C, O	Starter culture in different fermented foods and beverages.	[31,93]
	*Limosilactobacillus reuteri*	CY	Starter culture can produce antimicrobial molecules, such as organic acids, ethanol, and reuterin.	[94]
	*Lactobacillus delbruckii subsp. Lactis*	C, CY, Cf	Starter culture in yoghurt.	[95]
	*Lactococcus hircilactis*	Cf		
	*Leuconostoc fallax*	C, Cf	Starter culture for the production of butyric acid.	[96]
	*Leuconostoc mesenteroides*	C, CY, O		
	*Leuconostoc pseudomesenteroides*	C, Cf		
	*Streptococcus thermopjhilus*	CY	Starter culture in yoghurt and cheese by its rapidly growing in low pH conditions.	[97]
	*Streptococcus salivarius*	O		
	*Weissella cibaria*	C	High presence in different fermented foods; its redox potential influences the aromatic profile. Qualified Presumption of Safety (QPS) for food applications in in process.	[77]
	*Weissella fabaria*	C		
	*Weissella confusa*	O		
AAB	*Acetobacter cibinongensis*	C, Cf	Starter culture for food fermentation processes to favor the production of volatile compounds.	[98]
	*Acetobacter lovaniensis*	C		
	*Acetobacter malorum/cerevisiae*	C		
	*Acetobacter malorum/indonesiensis*	C, Cf		
	*Acetobacter fabarum*	C, Cf		
	*Acetobacter ghanensis*	C		
	*Acetobacter orientalis*	C, Cf		
	*Acetobacter okinawensis*	Cf		
	*Acetobacter pasteurianus*	C		
	*Acetobacter peroxydans*	C		
	*Acetobacter pomorum*	C		
	*Acetobacter senegalensis*	C, Cf		
	*Acetobacter syzygii*	C		
	*Acetobacter thaillandicus*	Cf		
	*Frateuria aurantia*	C	Starter culture potential when high concentrations of glucose are present.	[78]
	*Microbacterium lacticum*	C		
	*Gluconobacter cerevisiae*	Cf		
	*Gluconobacter oxydans*	C		
	*Gluconobacter sp.*	C		

* CJ = Chicha de Jora; Cf = Coffee; CM = Chicha de Morocho; C = Cacao; CH = Champús; CY = Chicha cassava; O = others; SC = Seven-grain Chicha.

## 4. Benefits and Risk 

The increased interest in fermented food products is mainly due to the benefits that some of the microorganisms provide to human health, including their contribution to a healthy gut microbiome and their potential role as probiotics, as well as the improved organoleptic characteristics of the fermented food products. Additionally, the enzymes produced by fermenting microorganisms, such as phytases, amilases, proteases, mannase, catalase, cellulose, pullulanase and lipases, help with the release of polyphenol compounds from vegetal matrixes [7,13,93,94,99,100]. 

However, the spontaneous nature of various fermentation processes may result in the presence of undesired fungi, such as *Aspergillus*, *Fusarium* and *Penicillium*, which are capable of producing toxic secondary metabolites, such as mycotoxins [101]. Approximately 400 mycotoxins have been reported in different cereals, grains, and other food, with aflatoxin, ochratoxin A, fumonisins, zearaleone, and patulin being amongst the most significant ones. Spontaneous fermented foods may carry traces of mycotoxins. Some researchers have reported aflatoxin and ochratoxin on fermented cacao bean clones [102], arabica and robusta coffee [103], and in the different grains used for the production of traditional fermented beverages [104]. Similarly, the corn and rice used for the production of different types of chicha can carry fungal species that may produce toxic metabolites [105,106]. The development of mycotoxins has been associated to the length of storage, excessive moisture, and unsanitary handling of certain fermented foods [107]. For instance, a recent study carried out in Brazil detected aflatoxin contamination in 38% of the analyzed cacao samples, while ochratoxin A was detected in 18% of the samples [103]. Similarly, high levels of ochratoxin A were detected in about 25% of the coffee samples [104]. In Ecuador, mycotoxins have also been detected in 23% of paddy rice, 33% of white wheat noodles, and 17% of oat flakes [104].

## 5. Future Perspectives 

Nowadays, there is a great variety of fermented foods produced spontaneously. However, with an increasing body of knowledge regarding the microbial community involved in the fermentation processes and technological advancements regarding large-scale fermentations, there is a need to select proper microbial strains that can steer the fermentation process, select quality raw materials, and apply proper process control. Additional research is needed for reducing the risk of contamination and negative changes in the sensory properties.

## 6. Development of Starter Cultures 

As microorganisms play an important role in food fermentation, there is a growing interest in the development of starter cultures. Commercial starter cultures of yeasts and bacteria are used for the production of bread, beer, wine, and cheese [108,109,110]. The commercially available starter cultures have been selected considering the ability of the strains to withstand strong stress conditions during the fermentation process, the identification of the key metabolites produced, and the evaluation of the necessary technological parameter needed for a proper fermentation process [94,110,111]. With the purpose of controlling and even steering the fermentation process, many studies have considered various combinations of microorganisms, among which are *Saccharomyces cerevisiae* in conjunction with other yeasts, such as *Pichia* and *Hanseniaspora*, for cacao and wine fermentation [84,85]. Also, different strains of *Lactobacillus* can be used as starter culture, due to their probiotic properties, lactic acid production, and biological reduction of mycotoxin-producing mold-fermented drinks [91,92]. 

## 7. Concluding Remarks

With the increasing demand of fermented foods, it becomes necessary to improve the productivity, food safety, and efficiency of the manufacturing processes, while decreasing production costs. Further research on innovative fermentation processes should include the bioprospection of microbial ecosystems for starter culture selection and development, as well as the natural inclusion of new flavors. Additionally, there is a need to control the intrinsic and extrinsic factors that may affect the fermentation processes. In general, the use of fermentors that allow monitoring the fermentation conditions, as well as the use of biosensors or similar devices for the control and discrimination of undesired microorganisms and sensitive detection of pathogenic bacteria, could allow to obtain a stable and safe final product [81,112].

Amongst the fermented foods mentioned in this review, cacao and coffee have been the most investigated products, given their economic importance. Further research is needed to better characterize the fermentation processes of other traditional fermented foods, such as chicha and champús. However, it is possible to apply the knowledge obtained from cacao and coffee research to understand the processes of other fermented foods. 

## Figures and Tables

**Figure 1 foods-11-01854-f001:**
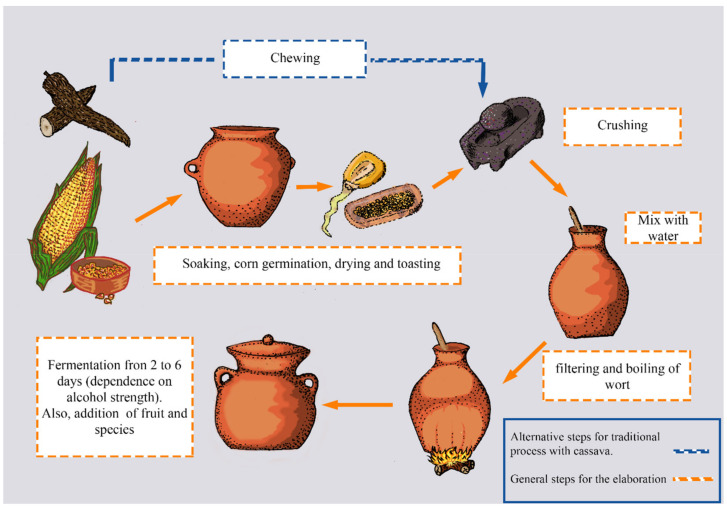
Flow diagram depicting the production process of chicha. Depending on the geographical location, cassava or corn is used as the main ingredient.

**Figure 2 foods-11-01854-f002:**
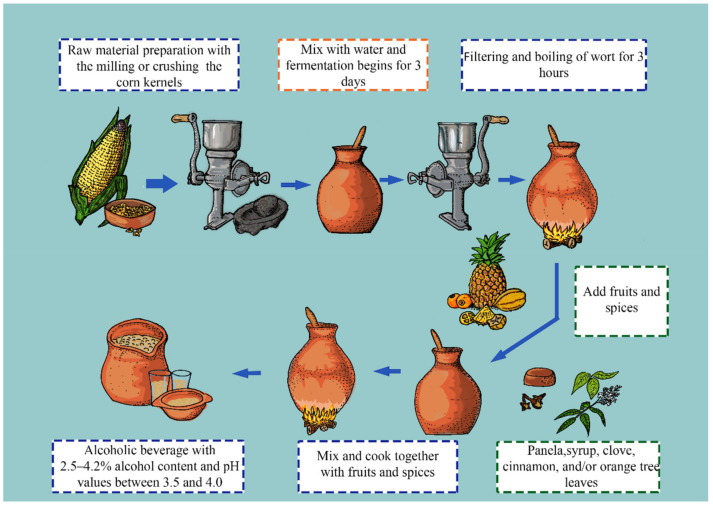
Flow diagram depicting the champús brewing process, considered as the indigenous Andean beer.

**Table 1 foods-11-01854-t001:** Diversity indices of fermented cacao and coffee.

Country	Food	Community	Diversity Indices			Reference
			Chao	Shannon	Simpson	
Brazil	Cacao	Bacteria	69.95	1.24	0.4	[40]
		Fungi	75.2	2.04	0.62	
Colombia	Cacao	Fungi	−	2.67	−	[41]
Colombia	Coffee	Bacteria	602.31	2.12	4.37	[42]
		Fungi	1740.48	2.25	3.28	

## Data Availability

Data is contained within the article.

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
