# Peer review of "Traditional Fermented Foods from Ecuador: A Review with a Focus on Microbial Diversity"

_foods, 2022, doi:10.3390/foods11131854_

Round 1

Reviewer 1 Report

The points I noticed in this manuscript are as follows.

Line 46: "acetc acid bacteria convert the carbohydrates mainly into organic acids such as lactic acid and acetic acid."

Acetic acid bacteria do not directly convert carbohydrates to acetic acid. Alcohol (ethanol) to acetic acid, or carbohydrate (glucose) to gluconic acid. Please reconsider this sentence.

Line 56:  A word "vinagar" is a misspelling of vinegar.

Table 2 should be separated by a horizontal line for each reference to help understanding.

Reviewer 2 Report

In general, the information presented by the authors is interesting, as they mention, that with the new trends, traditional fermented foods have begun to take great importance in the diet.

Some general comments I can mention:

I recommend re-reading the document, there are sentences where some words are repetitive, for example:

In lines 114-115: The biodiversity indexes have contributed to the analysis of fermented foods and have allowed an understanding of microbial ecosystems in the foods during the fermentation. In this case, delete "during the fermentation".

I find it difficult to understand the first two sections, cocoa, and coffee. The authors could improve the linkage of these two products and the relationship with fermented products, do you have fermented products based on coffee and cocoa?

L209: italics is the scientific name of the yeast.

L255-256. Information is repetitive with previous lines, also the idea is cut with the period and start the sentence with "And".

In general from lines 255-263. Rewrite the information and eliminate that which has already been mentioned in previous sections.

In Table 2, Is the strain of each microorganism available, since it is mentioned that some of them can be probiotic, however, this characteristic is strain-specific?

In section 4. Are there any studies on the potential risks of toxic compounds in the fermented foods mentioned in the study? Could the authors relate it to some other traditional fermented foods where the presence of these compounds has been reported?

In section 6, L313, "n", What is?
